# Effect of Minor Ce Substitution for Pr on the Glass Formability and Magnetocaloric Effect of a Fe$_{88}$Zr$_4$Pr$_4$B$_4$ Metallic Glass

Li-Ze Zhu [1], Qiang Wang [1], Shu-Hui Zheng [1], Peng-Jie Wang [1], Ding Ding [1,*], Ben-Zhen Tang [2], Peng Yu [2], Jin-Lei Yao [3] and Lei Xia [1,*]

1  Institute of Materials & Laboratory for Microstructure, Shanghai University, Shanghai 200072, China; lize_zhu@shu.edu.cn (L.-Z.Z.); mat_wq@shu.edu.cn (Q.W.); zhengshuhui@shu.edu.cn (S.-H.Z.); pengjie_w@shu.edu.cn (P.-J.W.)
2  Chongqing Key Laboratory of Photo-Electric Functional Materials, College of Physics and Electronic Engineering, Chongqing Normal University, Chongqing 401331, China; bz.tang@cqnu.edu.cn (B.-Z.T.); pengyu@cqnu.edu.cn (P.Y.)
3  Jiangsu Key Laboratory of Micro and Nano Heat Fluid Flow Technology and Energy Application, School of Physical Science and Technology, Suzhou University of Science and Technology, Suzhou 215009, China; jlyao@usts.edu.cn
*  Correspondence: d.ding@shu.edu.cn (D.D.); xialei@shu.edu.cn (L.X.); Tel.: +86-021-66135067 (D.D. & L.X.)

**Abstract:** In the present work, Fe$_{88}$Zr$_4$Pr$_3$B$_4$Ce$_1$ metallic glass (MG) was successfully prepared by minor Ce substitution for Pr, and compared with Fe$_{88}$Zr$_4$Pr$_4$B$_4$ MG in terms of glass forming ability (GFA), magnetic and magnetocaloric properties. The GFA, $T_c$ and the maximum magnetic entropy change ($-\Delta S_m^{peak}$) of the Fe$_{88}$Zr$_4$Pr$_3$B$_4$Ce$_1$ MG were found to decrease slightly. At the same time, the possible interaction mechanism of minor Ce replacing Pr was also explained. The critical exponents ($\beta$, $\gamma$ and n) obtained by the Kouvel–Fisher method indicate that Fe$_{88}$Zr$_4$Pr$_3$B$_4$Ce$_1$ MG near $T_c$ exhibits typical magnetocaloric behavior of fully amorphous alloys. The considerable maximum magnetic entropy change ($-\Delta S_m^{peak}$ = 3.84 J/(kg × K) under 5 T) near its Curie temperature ($T_c$ = 314 K) as well as RCP (~ 646.3 J/kg under 5 T) make the Fe$_{88}$Zr$_4$Pr$_3$B$_4$Ce$_1$ MG a better candidate as a component of the amorphous hybrids that exhibit table-shape magnetic entropy change profiles within the operation temperature interval of a magnetic refrigerator.

**Keywords:** metallic glass; glass forming ability; magnetic entropy change; adiabatic temperature change

## 1. Introduction

As is known, traditional gas compression–expansion refrigeration technology, dependent on fluorine-containing refrigerants, has many disadvantages such as greenhouse gas emission, destroying the ozone layer, low refrigeration efficiency, and so on. Therefore, magnetic refrigeration (MR) technology using magnetocaloric refrigerant has received a lot of attention because of its high efficiency, lower energy loss, environmental friendliness and structural compactness [1–6]. Magnetocaloric refrigerants are the magnetic materials that emit/absorb heat adiabatically when a magnetic field is applied/removed, which is called magnetocaloric effect (MCE) [4–6]. The magnetocaloric effect of the magnetic materials is induced by the reduction in magnetic entropy upon magnetization, and, as a consequence, the magnetocaloric properties of a magnet are usually evaluated by the change in magnetic entropy ($-\Delta S_m$) under a certain magnetic field. As such, early research focused on the MCE of first-order magnetic phase transition (FOMPT) materials that exhibit a sharp $-\Delta S_m$ profile with rather high maximum $-\Delta S_m$ ($-\Delta S_m^{peak}$) [7–9]. However, the narrow working temperature intervals of these FOMPT materials make them difficult to match the requirements of magnetic refrigerants working in an Ericsson cycle; that is, a fattened $-\Delta S_m$ curve over the range of operating temperatures in a magnetic refrigerator [10]. In

addition, the FOMPT materials inevitably show some disadvantages such as high magnetic and thermal hysteresis [11]. In contrast, amorphous magnetocaloric alloys that experience a second-order magnetic phase transition (SOMPT) exhibit several characteristics superior to the FOMPT materials, such as low energy loss induced by their negligible coercivity and high electric resistance, broadened $-\Delta S_m$ curve and tunable $-\Delta S_m$ peak temperature that make them easily composed to achieve the fattened $-\Delta S_m$ curve [12–31]. Unfortunately, although rare earth (RE)-based metallic glasses (MGs) exhibit rather high glass formability (GFA), excellent $-\Delta S_m^{peak}$ at low temperature and ultrahigh refrigeration capacity (RC), their formability and $-\Delta S_m^{peak}$ get worse when their Curie temperature ($T_c$) increases to or above the ambient temperature [12–18]. Thus, RE-based amorphous magnetocaloric alloys are more likely to be applied in low temperature refrigeration instead of room temperature (RT) refrigeration. The Fe-based amorphous magnetocaloric alloys exhibit good glass formability when their $T_c$ is near the ambient temperature, but their $-\Delta S_m^{peak}$ is usually very low [19–23]. For example, Fe-Zr-B MGs with $T_c$ ranging from the cold end ($T_{Cold}$) to the hot end ($T_{Hot}$) of domestic cooling equipment can be easily fabricated, but their $-\Delta S_m^{peak}$ under 5 T is less than 3.34 J/(kg $\times$ K) [21–23], which is far from enough for them to be utilized as cooling agents in domestic cooling appliances. More recently, by microalloying the Fe-Zr-B MGs with other transition metals or RE metals, we successfully adjusted the $T_c$ and improved the $-\Delta S_m^{peak}$ of the Fe-Zr-B MGs [24–31]. For instance, the $-\Delta S_m^{peak}$ under 5 T reaches 3.55 J/(kg $\times$ K) at 336 K in the $Fe_{85}Co_3Zr_5B_4Nb_3$ amorphous ribbon [24] and at 333 K in the $Fe_{85}Zr_8B_4Sm_3$ amorphous ribbon [25]; it reaches 4.0 J/(kg $\times$ K) at 323 K in the $Fe_{88}Zr_4Pr_4B_4$ amorphous ribbon [26] and 4.10 J/(kg $\times$ K) at 335 K in the amorphous $Fe_{88}Zr_4Nd_4B_4$ ribbon [27].

However, preliminary results show that the excellent MCE of the iron-based MGs appear near or above the $T_{Hot}$ of a domestic refrigerator. It is known that the high $-\Delta S_m^{peak}$ at temperatures higher than $T_{Cold}$ but lower than $T_{Hot}$ is also required for the construction of fattened $-\Delta S_m$ curves suitable for the Ericsson refrigeration cycle. Thus, it is necessary to develop a new type of metallic glass with excellent MCE at RT. As such, it is critical to decrease the $-\Delta S_m$ peak temperature of the iron-based MGs without dramatically deteriorating their $-\Delta S_m^{peak}$. In the present work, according to our preliminary results on the effect of Ce substitution on the $T_c$ and $-\Delta S_m^{peak}$ of the Fe-Zr-B amorphous alloys [32], we add minor Ce to replace the Pr element in the $Fe_{88}Zr_4Pr_4B_4$ amorphous alloy for the purpose of obtaining good magnetocaloric properties at a temperature slightly lower than the $T_{Hot}$ of a domestic refrigerator. The mechanism for the influence of minor Ce substitution on the magnetic as well as magnetocaloric properties of the $Fe_{88}Zr_4Pr_4B_4$ metallic glass was also investigated. The research results provide a feasible path for the Fe-Zr-B-RE amorphous alloy to reduce $T_c$ and avoid significant deterioration of magnetocaloric properties while reducing costs.

## 2. Materials and Methods

The $Fe_{88}Zr_4Pr_3B_4Ce_1$ ingot was manufactured by arc-melting the high purity raw materials more than five times to ensure uniformity of composition [33]. Ribbons were fabricated by spraying the $Fe_{88}Zr_4Pr_3B_4Ce_1$ melt from a quartz tube to the surface of a copper roller rotating at a linear velocity of 55 m/s. The whole sample preparation process is protected by a high purity Ar atmosphere. The cross-sectional morphology of the $Fe_{88}Zr_4Pr_3B_4Ce_1$ as-spun ribbon was characterized through a Hitachi tungsten filament scanning electron microscope (SEM, model SU-1500). The ~40-μm-thickness as-spun ribbons were selected for structural analysis by X-ray diffraction (XRD) using a Cu $K_\alpha$ radiation with a scanning speed of 1 $^\circ$/min on a PANnalytical spectrometer. Under program-controlled temperature conditions, the glass transition behavior, melting and crystallization of $Fe_{88}Zr_4Pr_3B_4Ce_1$ ribbons were distinguished by measuring the power difference between the sample and the reference material as a function of temperature (i.e., the thermal effect information related to heat absorption and release). Hence, the thermodynamic parameters, including glass transition temperature ($T_g$), initial crystallization temperature ($T_x$) and liquidus tempera-

ture ($T_l$), were derived from the differential scanning calorimetry (DSC) curve measured by a NETZSCH DSC-404 C calorimeter at a heating speed of 20 K/min to evaluate the formability of the MG ribbon. The temperature dependence of the heat capacity ($C_p(T)$) curve of the glassy sample was measured by a Perkin-Elmer DIAMOND calorimeter. The magnetic measurements of the amorphous ribbons, including magnetization vs. temperature (*M-T*) curve, isothermal magnetization (*M-H*) curve and hysteresis loop, were performed on the vibrating sample magnetometer (VSM) module of a physical property measurement system (PPMS, model 6000, Quantum Design) after applying an oscillating magnetic field to a fully amorphous ribbon to eliminate residual magnetism. The sample for magnetic measurement was prepared by sticking several ribbons together using non-magnetic cement. To minimize the impact of demagnetization, the magnetic field is applied parallel to the length of the sample.

## 3. Results and Discussion

The $Fe_{88}Zr_4Pr_3B_4Ce_1$ as-spun ribbon is amorphous according to its XRD pattern shown in Figure 1. The cross-sectional morphology (the upper left inset of Figure 1a) and the prepared samples (the upper right inset of Figure 1a) of the $Fe_{88}Zr_4Pr_3B_4Ce_1$ as-spun ribbon, indicate a thickness of ~40 μm and a width of ~2 mm. The glassy characteristic of the $Fe_{88}Zr_4Pr_3B_4Ce_1$ ribbon is also illustrated by the upward glass transition hump before the downward crystallization peak on its DSC trace, as shown in Figure 1b. The onset of $T_g$ and $T_x$ of the amorphous ribbon determined from its DSC trace is ~795 K and ~856 K, respectively. The $T_l$ of $Fe_{88}Zr_4Pr_3B_4Ce_1$ alloy obtained from its melting curve, which is illustrated in the inset of Figure 1b, is determined to be ~1545 K. Therefore, we can assess the GFA of the $Fe_{88}Zr_4Pr_3B_4Ce_1$ amorphous sample by calculating the reduced glass transition temperature ($T_{rg} = T_g/T_l = 0.515$) [34] as well as the parameter $\gamma$ ($= T_x/(T_g + T_l) = 0.366$) [35]. The $T_{rg}$ of the $Fe_{88}Zr_4Pr_3B_4Ce_1$ MG sample is slightly higher than that of the $Fe_{88}Zr_4Pr_4B_4$ MG [26], while the $\gamma$ parameter is slightly decreased by the Ce substitution. Therefore, it seems that the Ce addition does not obviously change the glass formability of the $Fe_{88}Zr_4Pr_4B_4$ metallic glass. On the other hand, although the $Fe_{88}Zr_4Pr_3B_4Ce_1$ as well as $Fe_{88}Zr_4Pr_4B_4$ MGs do not show $T_{rg}$ and $\gamma$ values comparable to the bulk metallic glasses, their $T_{rg}$ and $\gamma$ values are still larger than most other Fe-Zr-B MGs [21–23], indicating that the $Fe_{88}Zr_4Pr_3B_4Ce_1$ and $Fe_{88}Zr_4Pr_4B_4$ alloys can be easily prepared into MG ribbon.

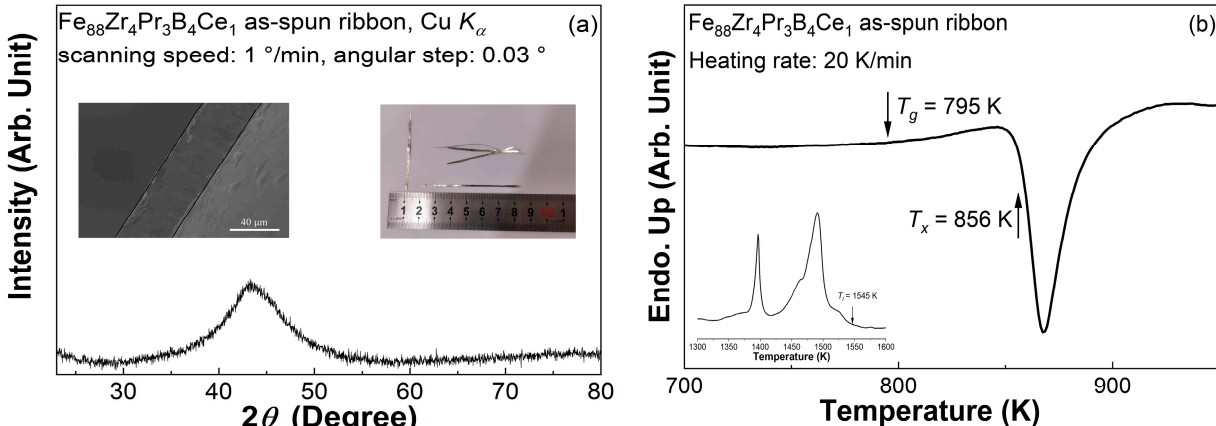

**Figure 1.** (**a**) XRD pattern of the $Fe_{88}Zr_4Pr_3B_4Ce_1$ as-spun ribbon measured at the scanning speed of 1 °/min: the upper-left is the cross-section morphology, the upper-right is the prepared sample; (**b**) The DSC traces and melting behaviors (inset) of the $Fe_{88}Zr_4Pr_3B_4Ce_1$ as-spun ribbon.

The *M-T* curve under 0.03 T of $Fe_{88}Zr_4Pr_3B_4Ce_1$ sample was measured after a zero-field-cooling process from RT. Figure 2a depicts the *M-T* curve under 0.03 T of the $Fe_{88}Zr_4Pr_3B_4Ce_1$ glassy sample. The ferromagnetic materials exhibit strong magnetism when magnetized.

However, as the temperature increases, the intensification of thermal motion will affect the ordered arrangement of magnetic moments of the magnetic domain. When the temperature reaches enough to disrupt the orderly arrangement of magnetic moments of the magnetic domain, the magnetic domain is disrupted, the average magnetic moment becomes zero and the magnetism of ferromagnetic materials disappears and becomes paramagnetic. As seen in the $(dM/dT)$-$T$ plots of the sample in the inset, $T_c$ of the $Fe_{88}Zr_4Pr_3B_4Ce_1$ MG is thus determined at the minimum value of the $dM/dT$ to be 314 K, which is about 9 K lower than that of the $Fe_{88}Zr_4Pr_4B_4$ MG [26]. The decreased $T_c$ caused by the replacement of Ce for Pr may be closely related to the antiferromagnetic coupling of the Ce atom with the Fe atom [32]. The hysteresis loops of the $Fe_{88}Zr_4Pr_3B_4Ce_1$ MG ribbon, as displayed in Figure 2b, suggest that the MG is paramagnetism at 380 K and soft magnetism at 200 K. The $Fe_{88}Zr_4Pr_3B_4Ce_1$ MG exhibits excellent soft magnetic properties with almost zero hysteresis and high magnetic susceptibility at 200 K, both of which are typical characteristics of fully amorphous alloys and are essential for magnetocaloric materials. The saturation magnetization ($M_s$) of the $Fe_{88}Zr_4Pr_3B_4Ce_1$ alloy (~129 $Am^2/kg$ at 200 K) is slightly lower than that of the $Fe_{88}Zr_4Pr_4B_4$ MG (~137 $Am^2/kg$ at 200 K [26]), indicating that the magnetocaloric properties of the $Fe_{88}Zr_4Pr_3B_4Ce_1$ MG may be not as high as $Fe_{88}Zr_4Pr_4B_4$ MG because both the $M_s$ and the $-\Delta S_m$ of the amorphous alloys are primarily determined by the ordering of their magnetic moments.

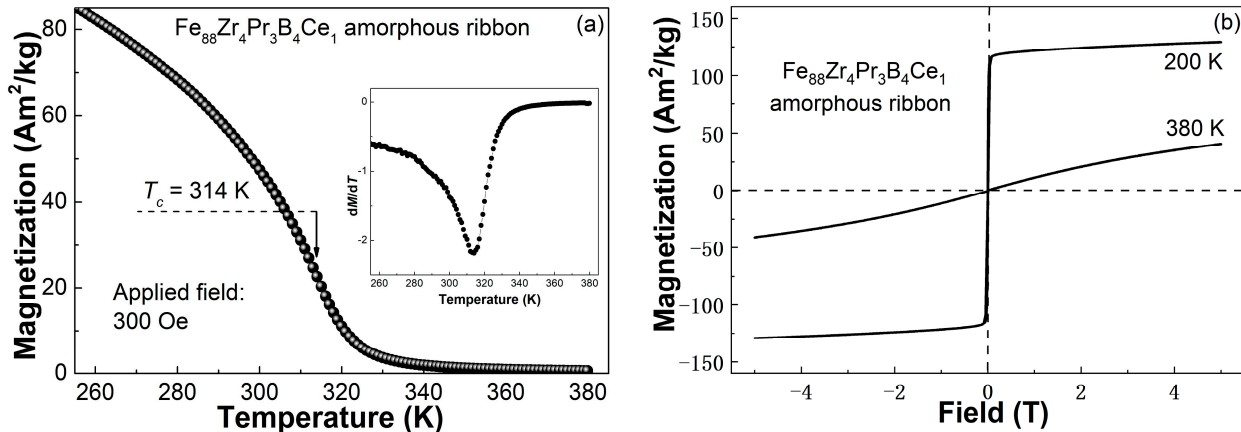

**Figure 2.** (**a**) $M$-$T$ curve of the $Fe_{88}Zr_4Pr_3B_4Ce_1$ amorphous ribbon measured under a field of 0.03 T, the inset is the $(dM/dT)$-$T$ curve; (**b**) Hysteresis loops of the $Fe_{88}Zr_4Pr_3B_4Ce_1$ amorphous ribbon measured at 200 K and 380 K under 5 T.

The temperature dependence of $-\Delta S_m$ ($-\Delta S_m$-$T$ curve) can be derived from the $M$-$H$ curves measured at various temperatures. Figure 3a displays the $M$-$H$ curves of the $Fe_{88}Zr_4Pr_3B_4Ce_1$ MG from 200 K to 380 K under 5 T. On the basis of these $M$-$H$ curves, the $M^2$-$H/M$ plots, namely the Arrott plots of the $Fe_{88}Zr_4Pr_3B_4Ce_1$ MG, can be established accordingly, as illustrated in Figure 3b. The Arrott plots ($M^2$-$H/M$) at each temperature show a positive slope and are almost parallel to each other from 200 K to 380 K, both of which indicate the typical feature of the materials experiencing a SOMPT according to the Banerjee criterion [36]. The second-order magnetic transition allows the alloy to undergo a continuous phase transition in a broad temperature range and hence leads to a better overall cooling capacity. The $-\Delta S_m$-$T$ curves under various external magnetic fields of the $Fe_{88}Zr_4Pr_3B_4Ce_1$ MG obtained from its $M$-$H$ curves are depicted in Figure 4a. The $-\Delta S_m^{peak}$ of the $Fe_{88}Zr_4Pr_3B_4Ce_1$ ribbon reaches 1.15 J/(kg × K) under 1 T, 1.57 J/(kg × K) under 1.5 T, 1.94 J/(kg × K) under 2 T, 2.63 J/(kg × K) under 3 T, 3.26 J/(kg × K) under 4 T and 3.84 J/(kg × K) under 5 T at 312.5 K. The $-\Delta S_m^{peak}$ of the $Fe_{88}Zr_4Pr_3B_4Ce_1$ ribbon is marginally lower than that of the $Fe_{88}Zr_4Pr_4B_4$ MG [26], probably because of the lower magnetic moment of the Ce atom than the Pr atom due to there being only one up-paired electron in the 4$f$ shell of Ce atom. The minor Ce atom sub-

stitution for Pr atom reduces the total magnetic moment of the $Fe_{88}Zr_4Pr_4B_4$ MG, which is confirmed by the effective magnetic moment ($\mu_{eff}$). As shown in Figure 4b, the temperature dependence of $H/M$ of the $Fe_{88}Zr_4Pr_4B_4$ and $Fe_{88}Zr_4Pr_3B_4Ce_1$ ribbons were obtained from their $M$-$T$ curves. According to the Curie–Weiss law [37], the slopes of the lines above their $T_c$ are correlated to the $\mu_{eff}$, and, thus, the $\mu_{eff}$ of the two MGs are calculated to be about 8.89 $\mu_B$ for $Fe_{88}Zr_4Pr_4B_4$ and 7.74 $\mu_B$ for $Fe_{88}Zr_4Pr_3B_4Ce_1$. Apparently, the $\mu_{eff}$ of the alloy is reduced with the addition of the Ce atom, resulting in a decrease in $-\Delta S_m^{peak}$.

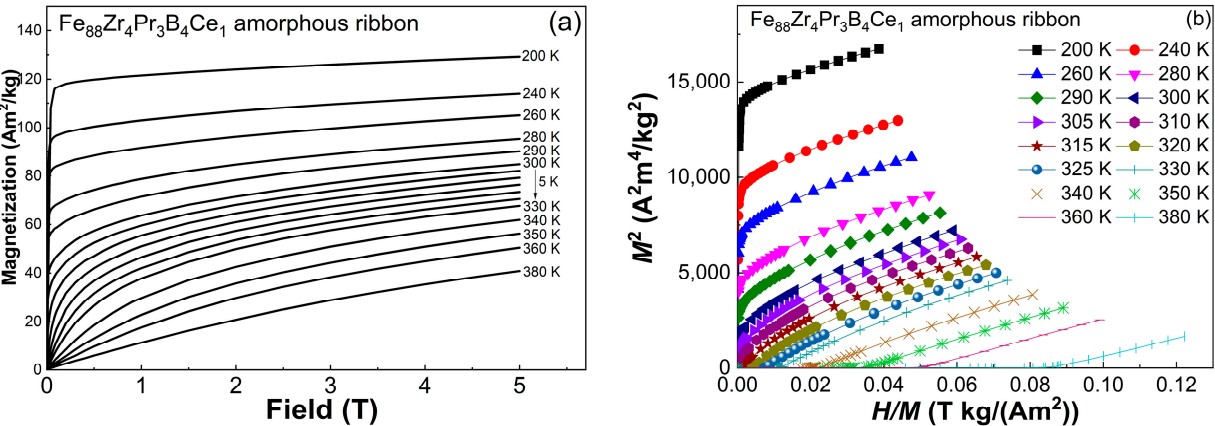

**Figure 3.** (**a**) Isothermal $M$-$H$ curves of the $Fe_{88}Zr_4Pr_3B_4Ce_1$ amorphous ribbon at various temperatures under 5 T; (**b**) Arrott plots of $Fe_{88}Zr_4Pr_3B_4Ce_1$ amorphous ribbon.

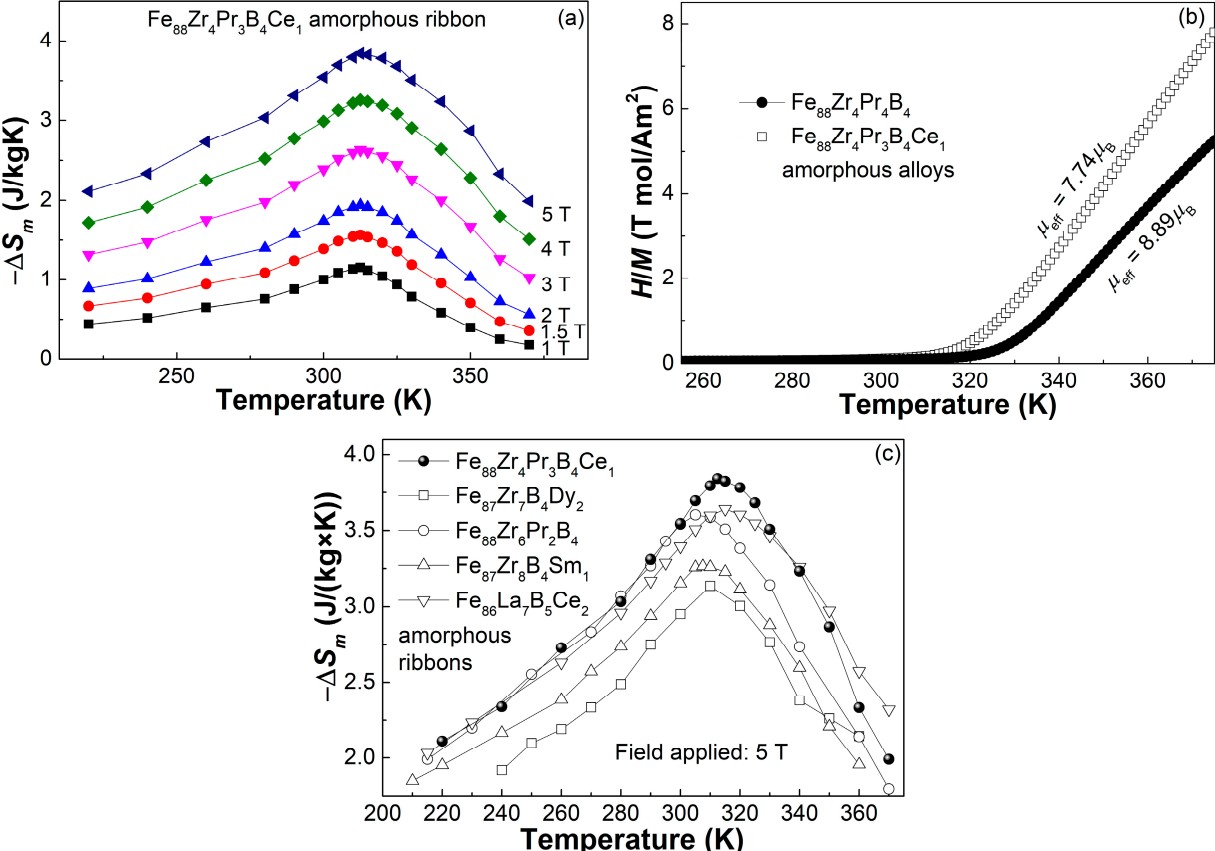

**Figure 4.** (**a**) $-\Delta S_m$-$T$ curves under various magnetic fields of the $Fe_{88}Zr_4Pr_3B_4Ce_1$ amorphous ribbon; (**b**) The effective magnetic moment of the $Fe_{88}Zr_4Pr_3B_4Ce_1$ and $Fe_{88}Zr_4Pr_4B_4$ amorphous ribbons; (**c**) $-\Delta S_m$-$T$ curves under 5 T of several Fe-based amorphous alloys with peak temperature near 310 K.

Although the $-\Delta S_m^{peak}$ of the $Fe_{88}Zr_4Pr_3B_4Ce_1$ ribbon is not as high as that of the $Fe_{88}Zr_4Pr_4B_4$ MG, it is still higher than the $-\Delta S_m^{peak}$ near 310 K of other amorphous alloys and even high entropy alloys (HEA) reported in the literature [25,26,38–41]. For example, its $-\Delta S_m^{peak}$ under 5 T is about 234% higher than that of the $Al_{20}Mn_{20}Fe_{20}Co_{15.5}Cr_{24.5}$ HEA (1.15 J/(kg × K) at 314 K [38]), 193% higher than that of the $Mn_{20}Al_{20}Co_{14}Fe_{23}Cr_{23}$ HEA (1.31 J/(kg × K) at 310 K [39]), 22.3% higher than that of the $Fe_{87}Zr_7B_4Dy_2$ MG (3.14 J/(kg × K) at 308 K [40]), 17.4% higher than that of the $Fe_{87}Zr_8B_4Sm_1$ MG (3.27 J/(kg × K) at 308 K [25]), 5.5% higher than that of the $Fe_{86}La_7B_5Ce_2$ MG (3.64 J/(kg × K) at 313 K [41]) and 6.67% larger than that of the $Fe_{88}Zr_6Pr_2B_4$ MG (3.6 J/(kg × K) at 306 K [26]). Figure 4c displays the $-\Delta S_m$-*T* curves of several iron-based MGs under 5 T. The $Fe_{88}Zr_4Pr_3B_4Ce_1$ MG ribbon shows a rather high $-\Delta S_m^{peak}$ near 310 K. On the other hand, the relative cooling power (*RCP* = $-\Delta S_m^{peak} \times \Delta T_{FWHM}$, where $\Delta T_{FWHM}$ is the full width at the half of $-\Delta S_m^{peak}$ [42]) of the $Fe_{88}Zr_4Pr_4B_3Ce_1$ MG, can be calculated as 164.7 J/kg under 1.5 T and 646.3 J/kg under 5 T according to the $-\Delta S_m$-*T* curve, both of which are similar to the values of amorphous alloys and much higher than those of the first-order magnetic transition alloys or compounds [26,41,43,44]. Since the $Fe_{88}Zr_4Pr_3B_4Ce_1$ MG experiences an SOMPT, it exhibits large value of magnetic entropy changes over a wide temperature range, which may be caused by the coupling interaction between RE-RE and RE-TM. Therefore, it can be predicted that $Fe_{88}Zr_4Pr_3B_4Ce_1$ MG has a good magnetocaloric effect over a large temperature range.

After constructing the ln($-\Delta S_m$)-ln(*H*) plots at each temperature, we can achieve their slopes (defined as *n*) by linearly fitting and thus, observe the magnetocaloric behaviors of the $Fe_{88}Zr_4Pr_3B_4Ce_1$ MG in more detail. Figure 5a represents the temperature dependence of *n* (*n*-*T* curve) of the $Fe_{88}Zr_4Pr_3B_4Ce_1$ amorphous ribbon. Similar to other amorphous alloys [21,22,24,26,27,40], the *n* of the $Fe_{88}Zr_4Pr_3B_4Ce_1$ MG is close to 1 at temperatures well below its $T_c$, and smoothly drops to the minimum value near its $T_c$, then dramatically increases with the increasing temperature and approaches 2 at temperatures much higher than its $T_c$. The minimum *n* value of the $Fe_{88}Zr_4Pr_3B_4Ce_1$ MG ribbon, which appears at 312.5 K and is shown in the inset of Figure 5a, is ~0.747 and is close to the predicted value of amorphous alloys proposed by V. Franco et al. based on the Arrott–Nokes equation [45]. Both the *n*-*T* curve and the minimum *n* value of the $Fe_{88}Zr_4Pr_3B_4Ce_1$ MG indicate typical magnetocaloric behaviors similar to those of fully MGs.

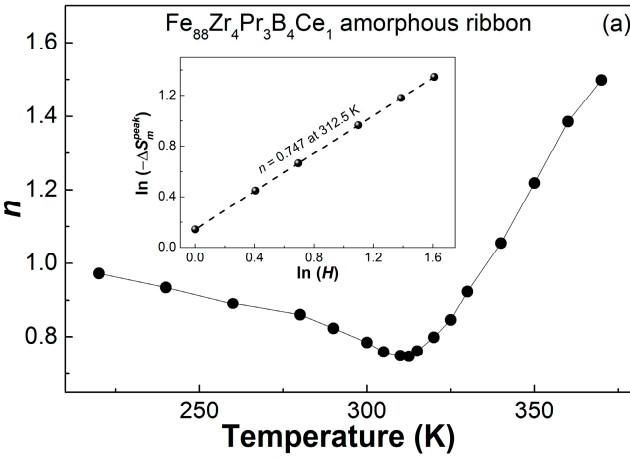

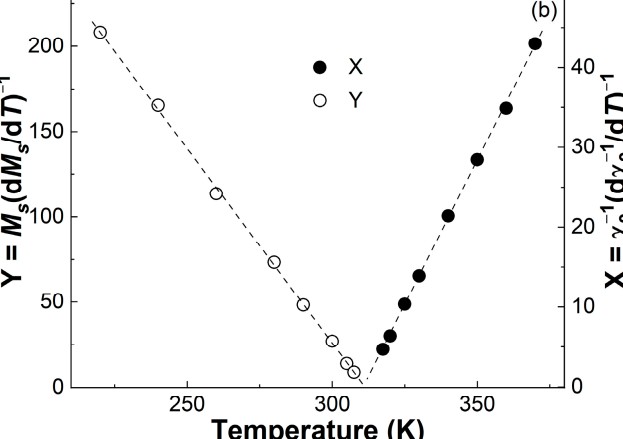

**Figure 5.** *Cont.*

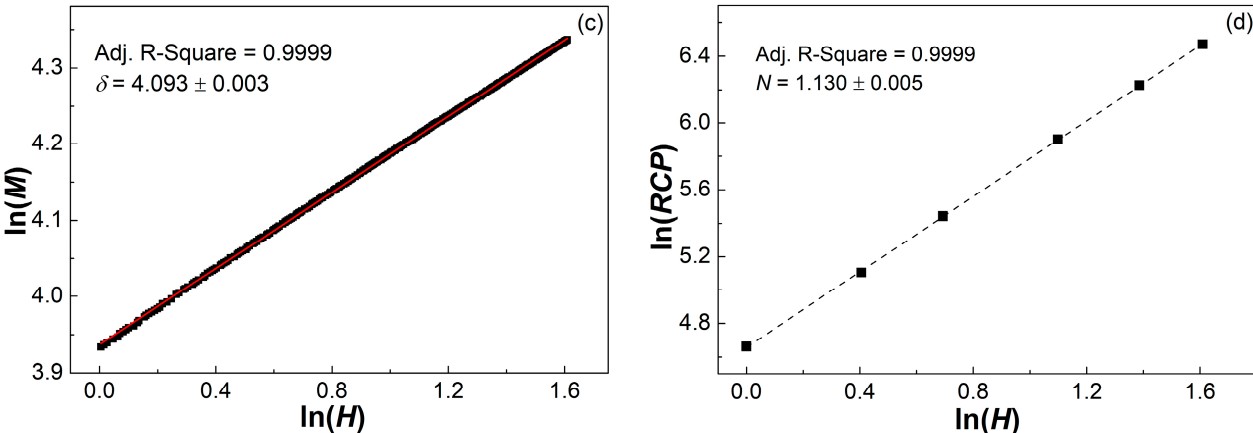

**Figure 5.** (**a**) The *n-T* curve of the $Fe_{88}Zr_4Pr_3B_4Ce_1$ amorphous ribbon, the inset is the linear fitting of the $\ln(-\Delta S_m^{peak})$ vs. $\ln(H)$ at 312.5 K; (**b**) Temperature dependence of $M_{st}(T)$ and $\chi_0(T)^{-1}$ of the $Fe_{88}Zr_4Pr_3B_4Ce_1$ amorphous ribbon; (**c**) The $\ln(M)$ vs. $\ln(H)$ plot at 315 K of the $Fe_{88}Zr_4Pr_3B_4Ce_1$ MG; (**d**) The $\ln(RCP)$-$\ln(H)$ plot of the $Fe_{88}Zr_4Pr_3B_4Ce_1$ MG.

On the other hand, the magnetocaloric behavior near its $T_c$ of the $Fe_{88}Zr_4Pr_3B_4Ce_1$ MG can also be explored by its critical exponents; that is, $n(T_c) = 1 + (\beta - 1)/(\beta + \gamma)$ [45]. Wherein, $\beta$ and $\gamma$ are the exponents related to spontaneous magnetization ($M_{st}$) and initial susceptibility ($\chi_0$), respectively, which can be described as follows [46]:

$$M_{st}(T) = M_0(-\varepsilon)^\beta, \varepsilon < 0, T < T_c \tag{1}$$

$$\chi_0(T)^{-1} = (H_0/M_0)\varepsilon^\gamma, \varepsilon > 0, T > T_c \tag{2}$$

where $M_0$ and $H_0$ are the critical amplitudes, $\varepsilon = (T - T_c)/T_c$ is the reduced temperature. Based on Equations (1) and (2), Kouvel and Fisher proposed a method to determine the critical exponents $\beta$ and $\gamma$ with high accuracy, namely the Kouvel–Fisher (KF) method [47]. Equations (1) and (2) can be rewritten as:

$$Y = M_{st}(T) \cdot (dM_{st}(T)/dT)^{-1} = (T - T_c)/\beta \tag{3}$$

$$X = \chi_0(T)^{-1} \cdot (d\chi_0(T)^{-1}/dT)^{-1} = (T - T_c)/\gamma \tag{4}$$

As such, we constructed the modified Arrott plots ($M^{2.5}$-$(H/M)^{0.75}$) at various temperatures of the $Fe_{88}Zr_4Pr_3B_4Ce_1$ MG and obtained the temperature dependence of $M_{st}$ and $\chi_0^{-1}$ from the intersections of the linear extrapolation of high field regions with $M^{2.5}$ and $(H/M)^{0.75}$ axes, respectively. Figure 5b shows the temperature dependence of $M_{st}(T) \cdot (dM_{st}(T)/dT)^{-1}$ and $\chi_0(T)^{-1} \cdot (d\chi_0(T)^{-1}/dT)^{-1}$ of the ribbon. The critical exponents $\beta$ and $\gamma$ can be determined to be 0.438 and 1.384 from the slope of the linear fitting of the two plots. The values of $\beta$ and $\gamma$ are close to those of other iron-based MGs [45,48]. Therefore, the *n* value near $T_c$ based on the KF method is calculated to be 0.692, which is slightly lower than the *n* value based on the Arrott–Nokes equation, but still higher than the theoretical value of the mean field model [45,49,50]. The reason for this may be related to the unique short-range ordered microstructure of MGs.

Furthermore, the field dependence of the refrigeration capacity is also controlled by the critical exponent, as follows [51]:

$$RC \propto H^N, N = 1 + \frac{1}{\delta} \tag{5}$$

where $\delta$ is the critical magnetization isotherm at $T_c$, that is:

$$M = DH^{\frac{1}{\delta}} \tag{6}$$

$D$ is the critical amplitude. According to the Widom scaling relation ($\delta = 1 + \gamma/\beta$) [52], the exponent $\delta$ can be determined to be 4.160. Moreover, the determination of the exponent $\delta$ can also be obtained by the modified Equation (6), as follows [46]:

$$\ln M = \ln D + \frac{1}{\delta} \ln H \tag{7}$$

Taking into account the measurement increments of 5 K in the $M$-$H$ curves from 300 K to 330 K and the $T_c$ of 314 K for the $Fe_{88}Zr_4Pr_3B_4Ce_1$ MG, the $M$-$H$ curve at 315 K was selected to construct the $\ln(M)$ vs. $\ln(H)$ plot, as shown in Figure 5c. The linear fitting is rather accurate, with a regression coefficient (Adj. R-Square) of up to 0.9999. The value of the exponent $\delta$ is derived from the slope of the linear fitting to be $4.093 \pm 0.003$, which is close to the result based on the Widom scaling relation. Therefore, according to Equation (5), we can obtain that $RC$ is roughly proportional to $H^{1.24}$. The $\ln(RCP)$-$\ln(H)$ plot of the $Fe_{88}Zr_4Pr_3B_4Ce_1$ MG constructed from its $RCP$ under different fields is illustrated in Figure 5d. The plot also fits well linearly and the slope is determined to be $1.130 \pm 0.005$. Clearly, the $N$ value obtained from the $\ln(RCP)$-$\ln(H)$ plot is slightly lower than that obtained from the KF method and the modified Equation (6), but these values are around the range of iron-based MGs. The deviation of $n$ and $N$ are supposed to be due to the error in obtaining $M_{st}$ and $\chi_0^{-1}$ from the modified Arrott plots.

A phenomenological universal behavior for the $-\Delta S_m$ of the SOMPT materials has been proposed by V. Franco et al. [53]. The $-\Delta S_m$-$T$ curves under all magnetic fields are normalized with their respective $-\Delta S_m^{peak}$; that is, $\Delta S'(T, H_{max}) = \Delta S_m(T, H_{max}) / \Delta S_m^{peak}(T, H_{max})$. The temperature axis is divided into upper and lower parts with $T_c$ as the boundary, and rescaled in different ways, as follows:

$$\theta = \begin{cases} -(T - T_c)/(T_{r1} - T_c), T \leq T_c \\ (T - T_c)/(T_{r2} - T_c), T > T_c \end{cases} \tag{8}$$

where $T_{r1}$ and $T_{r2}$ are the starting and ending temperatures corresponding to $\Delta T_{FWHM}$ under different magnetic fields, respectively. Figure 6a shows $\Delta S_m/\Delta S_m^{peak}$-$\theta$ curves under different magnetic fields of the $Fe_{88}Zr_4Pr_3B_4Ce_1$ MG. We can find that the normalized $\Delta S_m$ curves under each magnetic field can collapse onto the same universal curve. This uniformity indicates the typically magnetocaloric behavior of the SOMPT $Fe_{88}Zr_4Pr_3B_4Ce_1$ MG.

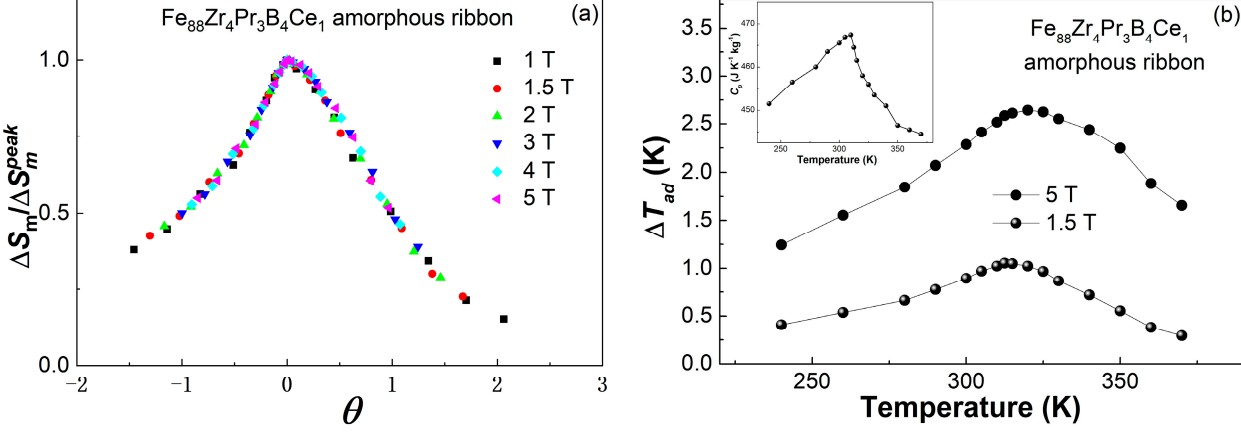

**Figure 6.** (**a**) Universal curves for $Fe_{88}Zr_4Pr_3B_4Ce_1$ amorphous ribbon under different magnetic fields; (**b**) $\Delta T_{ad}$-$T$ curves of the $Fe_{88}Zr_4Pr_3B_4Ce_1$ amorphous ribbon under 1.5 T and 5 T, the inset is its $C_p(T)$ curve.

To evaluate the magnetocaloric properties of the $Fe_{88}Zr_4Pr_3B_4Ce_1$ MG in a more direct way, we calculated the adiabatic temperature rise ($\Delta T_{ad}$) of the amorphous ribbon, as follows:

$$\Delta T_{ad}(T, 0 \rightarrow H) = -\frac{T}{C_p(T)}\Delta S_m(T, 0 \rightarrow H) \qquad (9)$$

Figure 6b illustrates the temperature dependence of $\Delta T_{ad}$ ($\Delta T_{ad}$-$T$ curve) of the $Fe_{88}Zr_4Pr_3B_4Ce_1$ MG obtained from its $-\Delta S_m$-$T$ curves and $C_p(T)$ curve (shown in the inset). The $\Delta T_{ad}$ reaches a maximum value of ~1.05 K under 1.5 T and ~2.64 K under 5 T, respectively. These well-known magnetocaloric materials (such as Gd [54], $Gd_5(Si_2Ge_2)$ [7], MnAs [8] and $Fe_{49}Rh_{51}$ [55]) undergo a first-order magnetic phase transition, exhibiting a giant magnetocaloric effect. Therefore, the magnetic entropy change curve shows extremely high sharp peaks within a narrow temperature range, and the magnetic entropy change peak value is much higher than that of the $Fe_{88}Zr_4Pr_3B_4Ce_1$ MG with a second-order magnetic phase transition in this study. Therefore, the RCP of $Fe_{88}Zr_4Pr_3B_4Ce_1$ MG is larger than that of famous magnetocaloric materials, but $\Delta T_{ad}$ is smaller than of these materials. Considering its relatively high $-\Delta S_m^{peak}$ at 312.5 K, RCP and $\Delta T_{ad}$, the $Fe_{88}Zr_4Pr_3B_4Ce_1$ MG is a prospective candidate for an intermediate component of magnetic refrigerants with a table-shape $-\Delta S_m$ curve within the interval between $T_{Cold}$ and $T_{Hot}$ of a domestic refrigerator.

Improving the $-\Delta S_m^{peak}$ near RT as much as possible seems to be an effective way to improve the magnetocaloric properties of Fe-based MGs. In the previous study of Fe-Ce-B ternary MGs [56], Ce was partially replaced by B, and it was found that with the decrease in Ce atoms, the antiferromagnetic coupling between Ce and Fe atoms was weakened, resulting in the enhancement of 3d-3d interaction between Fe atoms and thus $T_c$ increased. In the study of Fe-Zr-B ternary MGs [32], it was found that by replacing Zr with Ce, $T_c$ decreased from 306 K to 283 K. In summary, the composition dependence of the MCE in the Fe-Zr-Pr-B quaternary amorphous alloys system can be explained by the antiferromagnetic coupling between Ce and Fe atoms and the Ce-Pr interaction caused by the introduction of Ce atoms, which may lead to the weakening of the 3d-3d interaction between Fe atoms. Therefore, the substitution of Ce for Pr will reduce the $T_c$ of $Fe_{88}Zr_4Pr_4B_4$ MG. At the same time, because the magnetic moment of Ce atoms is lower than that of Pr atoms, the total magnetic entropy of the alloy will be reduced by the substitution of Ce for Pr, so the $-\Delta S_m^{peak}$ will be reduced. There are two reasons for choosing minor Ce to replace Pr in this research: firstly, Ce is cheaper than Pr, which can save costs; secondly, Ce and Pr are in adjacent positions in the periodic table of elements and the 4f shell is different from two electrons, which will not produce a large change. Based on the previous research results, it is expected that the total magnetic moment will not decrease too much while the minor Ce replaces Pr in reducing the $T_c$ of $Fe_{88}Zr_4Pr_4B_4$ MG, so it is more likely to obtain magnetocaloric materials with good magnetocaloric properties near RT.

## 4. Conclusions

In summary, minor Ce was selected to replace the Pr atom in a $Fe_{88}Zr_4Pr_4B_4$ MG, and the $Fe_{88}Zr_4Pr_3B_4Ce_1$ amorphous ribbon with a thickness of ~40 micrometer was successfully prepared. The influences of the minor Ce substitution for Pr on GFA, magnetic properties and magnetocaloric effect of the $Fe_{88}Zr_4Pr_4B_4$ MG, as well as their mechanisms, were further studied. The main conclusions are detailed below:

(i) The $T_{rg}$ and $\gamma$ indicate that the minor Ce substitution for Pr does not obviously change the glass formability of the $Fe_{88}Zr_4Pr_4B_4$ MG, but the glass formability of both the two ribbons is enough to vitrify them into glassy ribbon.

(ii) The $T_c$ of the $Fe_{88}Zr_4Pr_3B_4Ce_1$ ribbon decreases by 9 K compared with the $Fe_{88}Zr_4Pr_4B_4$ MG, which may be closely related to the antiferromagnetic coupling of the Ce atom with the Fe atom. The $Fe_{88}Zr_4Pr_3B_4Ce_1$ MG ribbon shows typical soft magnetic characteristics of fully amorphous alloys but slightly lower $M_s$ than that of the $Fe_{88}Zr_4Pr_4B_4$

MG at 200 K. The $M^2$-$H/M$ plots at various temperatures indicate the typical SOMPT feature of the $Fe_{88}Zr_4Pr_3B_4Ce_1$ MG.

(iii) According to the Maxwell Equation, the $-\Delta S_m^{peak}$ of the $Fe_{88}Zr_4Pr_3B_4Ce_1$ ribbon reaches 3.84 J/(kg $\times$ K) under 5 T at 312.5 K, which is slightly lower than that of the $Fe_{88}Zr_4Pr_4B_4$ MG but still higher than the $-\Delta S_m^{peak}$ near 310 K of other amorphous alloys and even high entropy alloys reported in literature.

(iv) The *n-T* curve, the minimum *n* value and the normalized universal curve of the $Fe_{88}Zr_4Pr_3B_4Ce_1$ MG ribbon also indicate the typical magnetocaloric behaviors of fully amorphous alloys. The values of *n* and *N* obtained by the KF method deviate slightly from those obtained by the linear fitting of the field dependence of $-\Delta S_m^{peak}$ and *RCP*, which may be due to the error in multiple derivation of the KF method.

Consequently, considering the relatively high $-\Delta S_m^{peak}$ near 310 K, RCP (~646.3 J/kg under 5 T) and $\Delta T_{ad}$ (~2.64 K under 5 T), the $Fe_{88}Zr_4Pr_3B_4Ce_1$ MG ribbon has great potential for application as an intermediate component of magnetic refrigerants with a flattened $-\Delta S_m$ curve in a domestic refrigerator.

**Author Contributions:** Conceptualization, L.X., P.Y. and J.-L.Y.; methodology, D.D.; validation, Q.W. and B.-Z.T.; investigation, L.-Z.Z., S.-H.Z. and P.-J.W.; data curation, Q.W., L.-Z.Z. and B.-Z.T.; writing—original draft preparation, L.-Z.Z. and Q.W.; writing—review and editing, D.D. and L.X.; funding acquisition, L.X. and P.Y. All authors have read and agreed to the published version of the manuscript.

**Funding:** This research was funded by the National Natural Science Foundation of China, grant number 51871139, 52071196 and 52071043.

**Data Availability Statement:** Data sharing is not applicable.

**Acknowledgments:** This research was technically supported by the Center for Advanced Microanalysis of Shanghai University.

**Conflicts of Interest:** The authors declare no conflict of interest.

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
