# Peer review of "Effect of Minor Ce Substitution for Pr on the Glass Formability and Magnetocaloric Effect of a Fe88Zr4Pr4B4 Metallic Glass"

_metals, doi:10.3390/met13091531_

Round 1

Reviewer 1 Report

The article is interesting and relevant, but needs to be revised.

1. Abstract 
- The abstract needs to be revised, especially the first part of it. At the moment the information in the abstract and conclusions are very similar, this aspect should be corrected. Please make appropriate changes.

2. Introduction
- It should clearly state the purpose of the research at the end of the introduction.

3. Materials and methods
- Reference should be made to the methodology of the manufacture of the ingot.
- It is necessary to expand the methodology of DSC and magnetic measurements. 

4. Figures
- Figure 1: the quality and readability for this figure needs to be improved. In addition, I pay attention to the authors that the inscription Cu2 Ka has been displaced in the figure. Please make appropriate changes.
- Figure 2: please improve the readability of the second graph in Figure 2a.
- Figure 6: please improve the readability of the second graph in Figure 6b.

5. Other comments 
- At the moment the article is a combination of methods without much deeper meaning. I recommend to the authors to make a new section Discussion, in which the authors would make a comparative analysis of previous experiments and compare the effectiveness of the obtained material. Why was Ce chosen? It would also be good to discuss this topic for deeper immersion of readers in the topic. 
- The conclusions need to be redone. At the moment the conclusions do not look favorable. I recommend making them point-by-point and clearly showing what the result is.
- It is necessary to align the position of equation 9.

Reviewer 2 Report

The authors investigated the effect of the Ce substitution for Pr on the glass formability, and magnetocaloric effect of a Fe88Zr4Pr4B4 metallic glass. This topic is original and relevant in the field. It is significant that large entropy change and RCP were observed in comparison with other magnetocaloric materials. The references are appropriate.

1. Fig.2 (a): As for the usual ferromagnetic materials, the magnetization suddenly decreases around TC in a weak magnetic field of the 0.03 T. As for this sample, the magnetization decreases in wide temperature range below 320 K. Explain the reason about this cause.

2. Fig. 4(a), (c) : Sm shows large value in a large temperature range. This indicates the magnetocaloric effect can be expected in a wide temperature range. Write down this fact in the main text.

3. L279-280 : The RCP under 5 T was larger than that of renowned magnetocaloric materials such as Gd, Gd5(Si2Ge2), MnAs, Fe49Rh51, etc. However, the deltaTad was smaller than that of these materials. Ex. Gd 5.7 K (under 2 T), Fe49Rh5112.9 K. Explain the reason why the deltaTad was smaller than others. 

Reviewer 3 Report

The article considers the properties of the Fe88Zr4Pr3B4Ce1 alloy in the form of metallic glass, as well as the presented results of changing the magnetic parameters in the case of variation of the components in the glass. In general, this line of research is very interesting and promising, since the presented results are of fundamental importance, which consists in studying the effect of changes in stoichiometry on the magnetic parameters of glasses, as well as practical significance, which lies in the further prospects for the use of Fe88Zr4Pr3B4Ce1 glasses in industry. The article corresponds to the subject of the declared journal, however, before making a final decision on this work, the authors should answer a number of questions that the reviewer had when reading it.

1. First, the choice of the Fe88Zr4Pr3B4Ce1 components, including the variation of the Pr and Ce components, should be explained in more detail and comparative analysis. In particular, the use of Ce in glasses usually causes a change in the optical properties due to its protective characteristics.

2. The authors should provide images of the obtained Fe88Zr4Pr3B4Ce1 glasses in order to visualize them and establish the possibility of using them as various light filters, if possible.

3. As a recommendation, the authors are invited to consider the possibility of using the Mössbauer spectroscopy method in order to establish hyperfine magnetic parameters, and as an addition, it would be good to use the energy dispersive analysis method to determine the isotropy of the distribution of elements in the composition of the obtained glasses.

4. The established effect of magnetic entropy depending on the composition of glasses should be described in more detail in view of the importance of this effect for practical applications.

5. The authors should give a short explanation of how exactly the glass thickness was determined and with what accuracy it was done, in other words, whether the thickness is maintained along the entire length of the tape.

6. Have the authors considered the possibility of using these glasses as shielding materials?

Round 2

Reviewer 1 Report

Thank you to the authors for the work done! The authors have corrected the comments and in my opinion the quality of the Manuscript has improved significantly. I recommend the article for publication. 

Reviewer 3 Report

The authors answered all the questions, the article can be accepted for publication.